# Learning Human Contribution Preferences in Collaborative Human-Robot Tasks

**Michelle Zhao**
Robotics Institute
Carnegie Mellon University United States
mzhao2@andrew.cmu.edu

**Reid Simmons**
Robotics Institute
Carnegie Mellon University United States
rsimmons@andrew.cmu.edu

**Henny Admoni**
Robotics Institute
Carnegie Mellon University United States
henny@cmu.edu

**Abstract:** In human-robot collaboration, both human and robotic agents must work together to achieve a set of shared objectives. However, each team member may have individual preferences, or constraints, for how they would like to contribute to the task. Effective teams align their actions to optimize task performance while satisfying each team member's constraints to the greatest extent possible. We propose a framework for representing human and robot contribution constraints in collaborative human-robot tasks. Additionally, we present an approach for learning a human partner's contribution constraint online during a collaborative interaction. We evaluate our approach using a variety of simulated human partners in a collaborative decluttering task. Our results demonstrate that our method improves team performance over baselines with some, but not all, simulated human partners. Furthermore, we conducted a pilot user study to gather preliminary insights into the effectiveness of our approach on task performance and collaborative fluency. Preliminary results suggest that pilot users performed fluently with our method, motivating further investigation into considering preferences that emerge from collaborative interactions.

**Keywords:** human-robot collaboration, reward learning, human-robot interaction

## 1 Introduction

As robots transition from machines in controlled environments to collaborative partners integrated into our daily lives, it is essential for them to align their behavior with our preferences and objectives. Human preferences can reflect constraints based on individual differences in capabilities [1, 2, 3], knowledge [4], intentions [5, 6], or preferred strategies for accomplishing shared goals [7, 8]. In interactions where the human's constraints dictate the team objectives, robots must align themselves to those constraints by learning and maximizing the objectives of the human [9, 10, 11, 12]. A crucial challenge in robot learning is to develop algorithms that can learn a reward function representing human preferences in collaborative human-robot tasks [13].

Robots also have a variety of task capabilities, requirements, and activity deadlines, which impose constraints on their behaviors [14, 15, 16]. These machine constraints require efficient allocation of appropriate tasks to appropriate agents, through adaptive scheduling [2, 3], or market-based methods [3, 17, 18]. Robots that are capable of self-assessment [19, 20, 21, 22] may even be able to identify constraints for their own actions. In real-world interactions, humans collaborating with robots must also need to align themselves to the constraints of the robot [23, 24].

Consider the task of collaboratively unloading a dishwasher with a robot. In this scenario, let's say the human partner does not trust the robot to handle fragile items, like glassware and china. However, they are comfortable allowing the robot to handle non-fragile objects, such as silverware. As such, the human's preferred contribution to the task is unloading glassware and china, while they would prefer the robot to unload non-fragile items. Similarly, the robot assesses that it can most

7th Conference on Robot Learning (CoRL 2023), Atlanta, USA.

safely unload plates due to their upright position in the dishwasher. An effective team strategy that plays to each team member's strengths [25] is for the robot to unload plates and silverware, while the human unloads glassware and other china. In order to achieve this strategy, the robot must learn the human's preference through interaction, and determine how to accomplish the task while maximally adhering to the human's preferences and its own constraints.

In this work, we capture both human and robot contribution constraints via a reward function based on the actions an agent performs towards some shared goal in a given state. There have been a variety of reward-based frameworks, including Cooperative Inverse Reinforcement Learning (CIRL) [12] which align robots to human constraints by modeling the human's constraints as a reward function the robot needs to learn and assist in maximizing [9, 10, 11]. We combine the notion of human contribution constraints with machine constraints and use a reward-based framework to represent a form of interaction: Interactive Contribution Preference Learning (ICPL), in which a robot partner must learn the human's contribution preferences under its own constraints and the objectives of the task. The robot and human operate within the same environment to complete a collaborative task under both the human's and robot's individual contribution reward functions. While the robot's individual reward is known to the team, the human's preferences must be learned by the robot. The observed human behavior, which provides insights into their preferences, is influenced by the robot's actions, its constraints, and the task objectives.

Our framework captures a novel type of collaboration in which successful teams maximize the task objectives while adhering to every partner's individual constraints. We propose a candidate approach to the collaborative task: Bayesian Learner (BaL). Our approach models the human as maximizing reward under the assumption that the robot will take the optimal action, setting expectations such that the robot will eventually learn the optimal action. This human model informs online Bayesian inverse reinforcement learning [26]. The robot then plans using the expected reward under updated beliefs. We also consider an extension of this method, where we add an information-gain term to the planning objective: Bayesian Information Seeking Learner (BaISL). BaISL then plans using the expected reward under updated beliefs towards states with high information gain potential given the actions available to the human. We evaluate BaL and BaISL in experiments with simulated human models and in a pilot user study with human partners. Our results indicate BaL and BaISL often improves team performance with some, but not all simulated human partners. The pilot results suggest that BaL and BaISL improves perceived collaborative fluency of human partners.

## 2    Related Work

**Human-Robot Collaboration**    Human-robot collaboration is the process in which humans and robots work together to achieve common goals [27, 13, 28]. Human-robot collaborations that are assistive in nature, like home service [29, 30] and healthcare robots [31, 32], align closely with the assistance paradigm proposed in [33], where the goals of the task are entirely defined by the human. Conversely, collaborations without rigid principal-assistant roles can allow the human and robot to differ in their approaches for achieving shared goals [24, 7]. For example, robots designed for collaborative manipulation of physical objects [28] or collaborative games [34, 35, 36] seek to contribute to shared goals, which are not necessarily dictated by human preferences. ICPL considers learning the human's preferences in the context of a collaborative task, where the objectives of the task are partially defined by the human's preferences.

**Value Alignment**    Value alignment in robotics refers to the design of collaborative agents that can learn and infer a human partner's objective and act towards achieving human's objective. Ensuring that robots have the correct reward function is an important problem in designing safe and reliable autonomous systems [37, 38, 9]. Existing approaches for value alignment focus on robot learning through interactions [12, 9], active learning of human preferences [39, 40], and teaching through demonstrations [41, 42]. Cooperative Inverse Reinforcement Learning (CIRL) [12] formulates value alignment as a two-player Markov game, in which human and robot must maximize a common reward function, but only the human knows the reward function. While the robot and human partner in CIRL are focused entirely on the task of ensuring the learning agent asymptotically converges to the same values as the human teacher, the robot and human in ICPL seek to balance multiple objectives. The actions taken by the human in ICPL are also in service of completing the collaborative task *and* take into account the robot's constraints.

**Reward Learning** A key part of value alignment and collaboration is understanding another agent's values, or utility functions, through observation of their actions [9, 43, 44, 45, 46, 6, 47, 48]. Inverse reinforcement learning (IRL) [49] infers an expert's underlying reward function from its observed behavior, typically in a single-agent setting [50, 51, 52]. Furthermore, when interacting with robots seeking to learn their reward functions, humans may convey their rewards through a number of different information modalities [53], including demonstrations [54], corrections [55], observations [56], and comparisons [39, 57]. Our reward learning occurs implicitly and online [58], where both agents must collaborate within the same two-agent MDP, where the robot agent lacks partial knowledge of the shared reward function.

## 3 Interactive Contribution Preference Learning

We formulate the problem of learning human contribution preferences subject to task objectives and individual constraints as a two-player Markov game. At a high level, an Interactive Contribution Preference Learning (ICPL) game involves both the robot and human operating simultaneously within the same environment to accomplish a known task objective. Each agent's behavior is additionally constrained by an individual reward function. The robot must learn a composite reward function that captures the **task reward**, **human individual reward**, and **robot individual reward** for the team in this collaborative task. While the human has full knowledge of the composite reward, known to the robot are only the **task reward** and **robot individual reward**. **Human individual reward** is unknown to the robot and is to be learned through interaction. An ICPL game is a two-player Markov game between a robot $r$, and a human, $h$, defined by the tuple $M = \langle \mathcal{S}, \{\mathcal{A}^r, \mathcal{A}^h\}, \mathcal{T}, \{\Theta^r, \Theta^h, \Theta^T\}, R_{\theta^T, \theta^r, \theta^h}, P_0, \gamma \rangle$.

- $\mathcal{S}$ the set of possible states: $s \in \mathcal{S}$
- $A^r$: action set available to the robot $r$: $a^r \in \mathcal{A}^r$
- $A^h$: action set available to the human $h$: $a^h \in \mathcal{A}^h$
- $\mathcal{T}(\cdot|\cdot, \cdot, \cdot)$: transition function, which is a conditional distribution on the next state, given previous state and action for both agents: $\mathcal{T}(s'|s, a^r, a^h)$
- $\Theta^T$: the space of possible **task reward** parameters.
- $\Theta^h$: the space of possible **human individual reward** parameters.
- $\Theta^r$: the space of possible **robot individual reward** parameters.
- $R_{\theta^T, \theta^r, \theta^h}(\cdot, \cdot, \cdot, \cdot) = C(R_{\theta^T}(\cdot, \cdot, \cdot, \cdot), R_{\theta^r}(\cdot, \cdot, \cdot), R_{\theta^h}(\cdot, \cdot, \cdot))$: the **composite reward** function that both players must maximize. The reward function is parameterized by $C$, a combination function of the individual and task reward functions. In our experiments, $C$ is the summation of the 3 reward components. $R_{\theta^T, \theta^r, \theta^h} : S \times \mathcal{A}^r \times \mathcal{A}^h \times S \times \Theta^T \to \mathbb{R}$. $R_{\theta^T}$ is known to both $h$ and $r$.
- $R_{\theta^T}(\cdot, \cdot, \cdot)$: a parameterized reward function representing how both players contribute to the group's task reward. $R_{\theta^T}$ represents both players' task-related payoffs, which we refer to as **task reward**. $R_{\theta^T} : S \times \mathcal{A}^r \times \mathcal{A}^h \times S \times \Theta^T \to \mathbb{R}$
- $R_{\theta^h}(\cdot, \cdot, \cdot)$: a parameterized reward function representing **human individual reward**, which represents the actions the human would prefer to perform in contribution to the task. $R_{\theta^h} : S \times \mathcal{A}^h \times S \times \Theta^h \to \mathbb{R}$. $R_{\theta^h}$ is known to only $h$.
- $R_{\theta^r}(\cdot, \cdot, \cdot)$: a parameterized reward function representing how $r$'s soft constraints enable it to contribute. $R_{\theta^r}$, or **robot individual reward**, represents how the robot $r$ is rewarded for the actions that it robot is able to take. The robot's capabilities may dictate these soft constraints. $R_{\theta^r} : S \times \mathcal{A}^r \times S \times \Theta^r \to \mathbb{R}$. $R_{\theta^r}$ is known to both $h$ and $r$.
- $P_0(\cdot, \cdot, \cdot, \cdot)$: initial state distribution. $P_0(s_0, \theta^T, \theta^r, \theta^h)$.
- $\gamma$: a discount factor $\gamma \in [0, 1]$

The game begins in an initial state $(s, \theta^r, \theta^h, \theta^T)$ sampled from $P_0$. Robot $r$ observes $\theta^r, \theta^T$, and human $h$ observes $\theta^h, \theta^r, \theta^T$. At each timestep $t$, robot $r$ and human $h$ observe the current state $s_t$ and select their actions $a_t^r, a_t^h$ according to policies $\pi^r$ and $\pi^h$. The environment transitions to the next state $s_{t+1} \sim \mathcal{T}(s'|s, a^r, a^h)$. The team receives, but does not observe, composite reward $R_{\theta^T, \theta^r, \theta^h}(s_t, a_t^r, a_t^h, s_{t+1}) = R_{\theta^T}(s_t, a_t^r, a_t^h, s_{t+1}) + R_{\theta^r}(s_t, a_t^r, s_{t+1}) + R_{\theta^h}(s_t, a_t^h, s_{t+1})$. The two agents observe each other's action selection. The process continues until the team reaches a

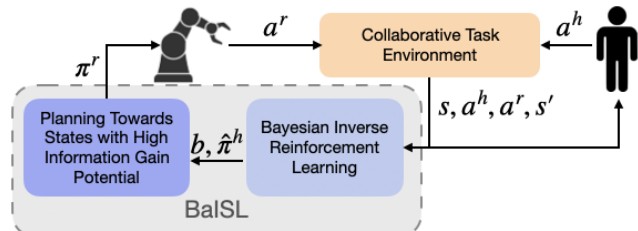

Figure 1: Upon observing the action of the human, current state, and next state, the robot updates its beliefs ($b$) over the hypothesis space ($\Theta^h$) and produces a predicted policy for the human ($\hat{\pi}^h$). The robot then replans its actions seeking information gain.

terminal state, $s_F$, where the final reward for the team $R_F = \sum_{i=0}^{F} R_{\theta^T, \theta^r, \theta^h}(s_i, a_i^r, a_i^h, s_{i+1})$ is received. The team only observes the total reward at the end of the interaction.

## 4 Approach: Bayesian Learner (BaL) and Bayesian Information Seeking Learner (BaISL)

We propose an algorithm for the robot in ICPL, called Bayesian Learner (BaL), and an extension of the algorithm called Bayesian Information Seeking Learner (BaISL). BaL consists of two primary online steps: (1) Bayesian Inverse Reinforcement Learning (BIRL) [26] using a human model with optimistic expectations of the robot, (2) planning given expected reward under beliefs. BaISL consists of the same two steps, but plans under beliefs towards informative states (Fig. 1). Implementation details in Appendix.

**Human Model for Bayesian Inverse Reinforcement Learning.** The robot estimates a belief, $b$, over the hypothesis space $\Theta^h$ of possible human individual rewards. In order to update its beliefs, the robot relies on a model of human decision-making in order to evaluate the likelihood of human actions. We model an *optimistic reward human* as a noisily rational actor who selects actions based on the expectation that the robot will take the optimal complementary action to maximize the team's reward. By understanding these expectations, the robot is able to deduce which human individual reward function would lead to such expectations. We also adopt the Boltzmann rational model of human behavior [59, 60], which models that the probability of choosing a particular option increases exponentially as its utility, or reward, increases compared to other available options. Over $t$ timesteps, the robot observes a trajectory of human actions, $\tau_t = \{(s_1, a_1^h, s_2), (s_2, a_2^h, s_3), ..., (s_t, a_t^h, s_{t+1})\}$. Since the human has full knowledge of the composite reward, we assume the human's policy is a stationary expert policy and can make the following independence assumption for all candidate human reward functions $\hat{\theta}^h \in \Theta^h$ (Eq. 10).

$$b(\hat{\theta}^h) = P(\hat{\theta}^h|\tau_t) \propto P((s_1, a_1^h)|\hat{\theta}^h)P((s_2, a_2^h)|\hat{\theta}^h)...P((s_t, a_t^h)|\hat{\theta}^h)P(\hat{\theta}^h) \tag{1}$$

Best-case reward, $r_{bc}$, is the reward $R_{\theta^T, \theta^r, \hat{\theta}^h}$ the team would receive, computed with candidate $\hat{\theta}^h$, if the robot had taken the action that would maximize the composite reward (Eq. 2).

$$r_{bc}(s_t, a_t^h|\hat{\theta}^h) = \max_{a^r \in \mathcal{A}^r} \sum_{s_{t+1}} R_{\theta^T, \theta^r, \hat{\theta}^h}(s_t, a_t^h, s_{t+1})\mathcal{T}(s_{t+1}|s, a_t^h, a_t^r) \tag{2}$$

Under the Boltzmann model, $r_{bc}(s_t, a_t^h|\hat{\theta}^h)$ serves as a potential function (Eq. 3), where $\beta$ is a rationality coefficient. The BIRL update yields a predicted human policy over all beliefs is $\hat{\pi}^h(a^h|s) = \sum_{\hat{\theta}^h} b(\hat{\theta}^h)P((s, a^h)|\hat{\theta}^h)$.

$$P((s_t, a_t^h)|\hat{\theta}^h) \propto \exp(\beta \cdot r_{bc}(s_t, a_t^h|\hat{\theta}^h)) \tag{3}$$

**Planning Under Expected Reward (BaL).** BaL robot acts according to the resulting policy $\pi^r$, where $Q(s, a^r, a^h, b)$ represents expected discounted future composite team rewards given $b$ (Eq 5).

$$\pi^r(s) \leftarrow \max_{a^r \in \mathcal{A}^r} \sum_{a^h \in \mathcal{A}^h} \sum_{\hat{\theta}^h} b(\hat{\theta}^h)\hat{\pi}^h(a^h|s; \hat{\theta}^h)\left(Q(s, a^r, a^h, b)\right) \tag{4}$$

**Planning Towards States with High Information Gain Potential (BaISL).** We consider an extension to BaL where we leverage the key insight that the robot's actions in a current state will affect the available actions of the human in the next state, which will in turn affect the human's ability to provide informative actions. Hence, the robot should seek to maximize team reward under its

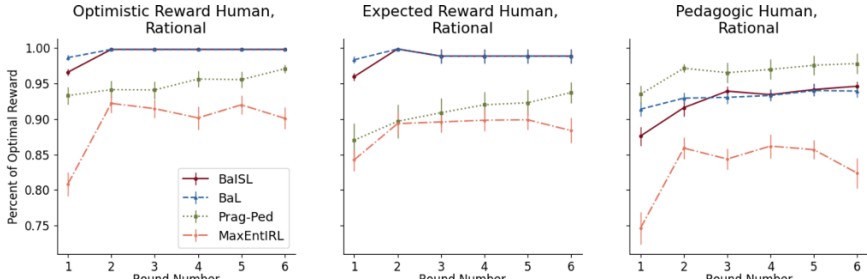

Figure 2: The human models are fully rational with $\beta = \infty$. Though BaISL's information seeking incurs cost during the first round, BaISL and BaL perform optimally in later rounds with *optimistic reward human* and the *expected reward human*. When paired with the *pedagogic human*, Prag-Ped outperforms the other three methods, due to the human model being the one Prag-Ped expects. BaISL and BaL perform similarly with the *pedagogic human*. Error bars represent standard error.

beliefs while seeking to allow the human to provide informative demonstrations or actions. First, we perform a Bellman backup to compute a task-based $Q$ given the current beliefs $b$ (Eq. 5).

$$Q(s, a^r, a^h, b) = \sum_{\hat{\theta}^h} b(\hat{\theta}^h) \sum_{s' \in S} T(s'|s, a^r, a^h) \left( R_{\hat{\theta}^h, \theta^r, \theta^T}(s, a^r, a^h, s') + \gamma V(s') \right) \quad (5)$$

We want some measure of information gain $I(b, s, a^r)$ the robot would receive if it takes action $a^r$ in state $s$ with the current set of beliefs $b$. $H(b)$ represents the entropy of the current beliefs $b$ over $\Theta^h$. We can easily measure the entropy of the new beliefs, $H(b|(s, a^h))$, given the human takes action $a^h$ in state $s$. We compute this value by updating the beliefs using the likelihood in Eq. 3, which yields updated belief $b'$ and estimated human policy $\hat{\pi}^h$. Then, the conditional entropy $H(b|(s, a^h))$ is measured by computing the entropy over $b'$, $H(b|(s, a^h)) = H(b')$. Since we want to evaluate the information gain for $a^r$ instead of $a^h$, we need some measure of $H(b|s, a^r)$, where $I(b, s, a^r) = H(b) - H(b|s, a^r)$. However, the robot and human act simultaneously, so the human will not be impacted by the action of the robot until the next timestep, once both actions of the human and robot are revealed and the environment transitions to the next state. Thus, the robot evaluates information gain potential, as the info gain achieved by the human giving the best entropy-minimizing action in the next state.

$$\mathbb{E}_{f,T}[H(b|s, a^r)] = \sum_{a^h \in \mathcal{A}^h} \sum_{s' \in \mathcal{S}} \mathcal{T}(s'|s, a^r, a^h) \hat{\pi}^h(a^h|s) H_{bc}(s', s, a^r, a^h) \quad (6)$$

$$H_{bc}(s', s, a^r, a^h) = \min_{a^{h'} \in \mathcal{A}^h} H(b|(s', a^{h'})) \quad (7)$$

We compute expected entropy using Equation 6. $H_{bc}$ represents the best-case entropy achieved from updating the beliefs given the human takes the most entropy-minimizing action in the next state (Eq. 7). It's important to note that the robot does not necessarily expect the human to always take the most entropy-minimizing action in every state; the robot simply seeks states that would provide the human the opportunity to give more informative decisions. Subsequently, the information gain of $a^r$ in state $s$ is added to the Q-value as a boost to actions that would guide the team towards states with the highest potential info gain (Eq. 8), where $I(b, s, a^r) = H(b) - \mathbb{E}_{\hat{\pi}^h, T}[H(b|s, a^r)]$. The $\alpha$ term (Eq. 9) is a switch that turns off the information gain boost when the hypotheses with maximum weight converge to predicting the same human action. BaISL robot acts according $\pi^r$:

$$\pi^r(s) \leftarrow \max_{a^r \in \mathcal{A}^r} \sum_{a^h \in \mathcal{A}^h} \sum_{\hat{\theta}^h} b(\hat{\theta}^h) \hat{\pi}^h(a^h|s; \hat{\theta}^h) \left( Q(s, a^r, a^h, b) + \alpha I(b, s, a^r) \right) \quad (8)$$

$$\alpha = | \bigcap_{\hat{\theta}^h \in \Theta^h} \{ a \mid \pi^h(a|s; \hat{\theta}^h) = \max_b \hat{\pi}^h(b|s; \hat{\theta}^h) \wedge b(\hat{\theta}^h) = \max_{\bar{\theta}^h} b(\bar{\theta}^h) \} | \quad (9)$$

## 5 Simulated Evaluation

**Task Scenario.** We evaluate our approach on a simulated collaborative task, with a series of simulated human partners. The human-robot team must clean up objects from a shared workspace. There exist $K = 3$ object types, where objects differ by fragility and weight. The three types of objects are (1) heavy and fragile, (2) heavy and non-fragile, and (3) light and fragile. The action space available to each player is {cleanup (heavy, fragile) object, cleanup (heavy, non-fragile) object, cleanup

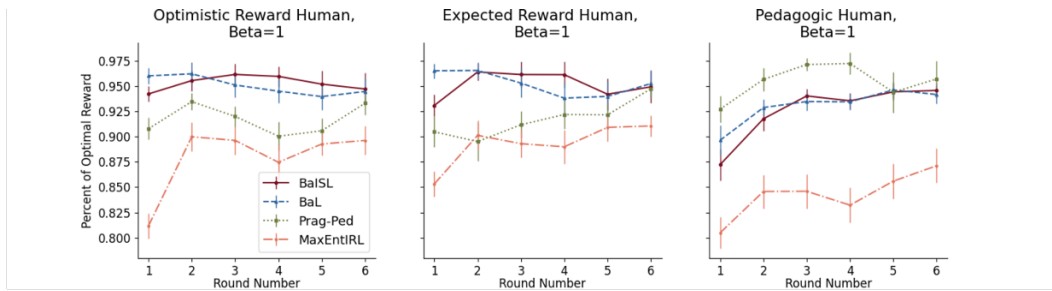

Figure 3: Boltzmann $\beta = 1$. With all 3 human models, BaISL incurs cost due to the information gain objective during the first two rounds but performs similarly or marginally better than BaL in later rounds. Only when paired with the *pedagogic human*, Prag-Ped outperforms the other methods.

(light, fragile) object, No-Op}. The heavy objects require both agents to jointly select the same action. Since the task objective is to declutter all objects, task reward is $+2$ for each object, with a $-1$ step cost. The human's individual reward, $\theta^h$, for each object type is randomly sampled from $\mathcal{U}(1, 10)$. The human's individual reward $\theta^h$ is observed only by the human. However, the robot has does access to $\Theta^h$: the hypothesis space for the human individual reward, which is the set of all permutations of $\theta^h$ values. The robot's individual reward $\theta^r$ is a random permutation of the same set of four reward values in $\theta^h$, and is known to both agents. In each instance of this game, the human and robot collaborate over $N = 6$ interactions. At the start of each interaction, there exists one object of each type to be decluttered. When all objects have been cleaned up, the interaction ends and the team receives the total composite reward achieved.

**Performance Metric.** We ran 100 random configurations of individual rewards in this collaborative task. We solve the full-information MDP with joint actions to obtain the optimal composite reward. Performance is measured by the percentage of optimal reward achieved in the final $N$th round.

**Baselines.** We compare our methods BaL (an ablation of BaISL) and BaISL to an alternative learning approach: Maximum Entropy Inverse Reinforcement Learning (MaxEnt IRL)[50], which approximates the composite reward, bypassing the need for individual components, while maintaining the same value-based planning method. This baseline examines the efficacy of separating out the three reward components versus learning a single joint reward function to approximate the composite. Lastly, we consider a second baseline: Pragmatic-Pedagogic Value Alignment (Prag-Ped)[10], a solution to CIRL in which the robot policy assumes that the human will act pedagogically accounting for the robot's beliefs. Technical details can be found in the Appendix.

**Simulated Human Partner.** We experiment with three types of simulated humans. For each agent type, we experiment with temperatures $\beta \to \infty$ (rational), and the other with $\beta = 1$ (noisy). (1) The *optimistic reward human* selects actions that maximize composite reward assuming the human takes the optimal complementary action. (2) The *expected reward human* selects actions that maximize *expected* composite reward under a uniform probability over all robot actions. (3) The *pedagogic human* uses the Prag-Ped solution in Fisac et al. [10] to act pedagogically by rolling out the robot's belief update.

**Results.** Our experiments with the noiseless, rational simulated human models ($\beta = \infty$) demonstrate that the robot policies which pair with human models that match their prediction of human behavior outperform the misaligned human-robot teams. BaISL, BaL, and Prag-Ped require that the robot assume some model of human behavior in order to define the likelihood function for human actions. BaISL and BaL model the human as being of the type *optimistic reward human*. Prag-Ped models the human as being of the type *pedagogic human*. Naturally, when BaISL and BaL pair with the *optimistic reward human*, BaISL and BaL have the correct model of the human, giving these approaches an advantage over Prag-Ped. Likewise, when Prag-Ped robot pairs with *pedagogic human*, Prag-Ped has the advantage of using the correct human model to make belief updates. Because of this advantage with the Pedagogic human, BaISL and BaL do not perform as well as Prag-Ped.

To combat the advantage of an approach having the correct human model, we include a human partner (Expected Reward human) which none of the robot approaches use as their human model. Importantly, with this third human type, we still that our approach performs well and outperforms the

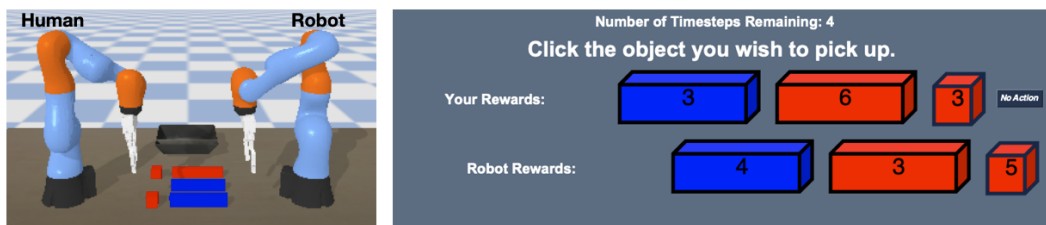

Figure 4: The user study interface requires the human to control one of two manipulators. The robot partner is the manipulator on the right. Users select actions at the object level using an interface.

baselines. BaISL and BaL outperform Prag-Ped with the rational, *expected reward human*, achieving optimal performance. As the matching would lead us to expect, the Prag-Ped robot outperforms BaISL and BaL with the rational, *pedagogic human* (Fig 2).

In our second set of experiments, $\beta = 1$ simulated humans make more noisy decisions, causing the robot's beliefs to become prone to erroneous updates, which may lead to worse performance. BaISL and BaL both perform well with the *optimistic reward human* and *expected reward human*. The approaches perform similarly to each other but not as well as Prag-Ped when paired with the *pedagogic human*. Generally, the BaISL robot performs more information-seeking actions, which are sometimes No-Op actions, early on during the first round. This tends to lead to lower performance in the first round than BaL, which immediately seeks to maximize expected reward (Fig 3). We find that our method with the information seeking term (BaISL) does not significantly improve performance over our method without the information seeking term (BaL). While not reflected in all randomly sample task configurations, when the rewards induce the human and robot to share the same preferred object, but the human's reward for picking up an item is higher than the robot's, we observe that our method is beneficial (Fig 6 in Appendix).

## 6  User Study

We ran a pilot user study, with five participants, to compare the collaboration of human partners with our BaL and BaISL agent against the two baselines. We investigated the impact of the independent variable: agent type, on dependent variables: task performance, and collaborative fluency. Task performance is defined by the percentage of optimal reward achieved. We measured four collaborative fluency variables [61]: overall fluency, the improvement of fluency over time, affinity towards the robot as a collaborative partner, and the perceived ability of the robot to learn the correct human individual reward, using four 5-point Likert scale questions whose values we aggregated in order to treat as a continuous variable [62]. As with the simulated evaluation, we evaluate BaL and BaISL against Prag-Ped and MaxEnt IRL. Each participant performed an ICPL task four times, once with each agent. The ICPL task was a decluttering task similar to the simulated task scenario, with $K = 3$ object types, where objects differ by color (representing fragility) and weight (Fig 4). Each of 3 rounds began with 1-2 objects of each type, where the starting configuration of objects was identical per round. After each task, participants were asked a series of survey questions. Participants were given two scaffolded training tasks to familiarize themselves with the online game's controls.

While we did not perform statistical analyses due to the small sample size, we found general trends indicating that users performed closest to optimally when working with the BaISL and BaL robots. Teams with BaISL robot were able to more consistently maintain close to optimal rewards over the three rounds. On average, in the third and final round, the BaISL teams achieved 86% of the total reward, while BaL teams achieved 72%, Prag-Ped teams achieved 48%, and MaxEnt IRL teams achieved 62%. This indicates that the *optimistic reward human* model might more closely resemble human reasoning in this collaborative task than the *pedagogic human* model. The MaxEnt baseline suffers in the first round because it relies on successful demonstrations in order to begin learning the reward. Its performance improves throughout the three rounds but never reaches the performance of BaISL and BaL. Participants also self-reported higher collaborative fluency with the BaISL and BaL models than the other two baselines. They also perceived the largest improvement in fluency over time with BaISL and BaL, with BaISL slightly higher than BaL. This may be related to our expectation that the BaISL model begins by taking information-seeking actions, rather than

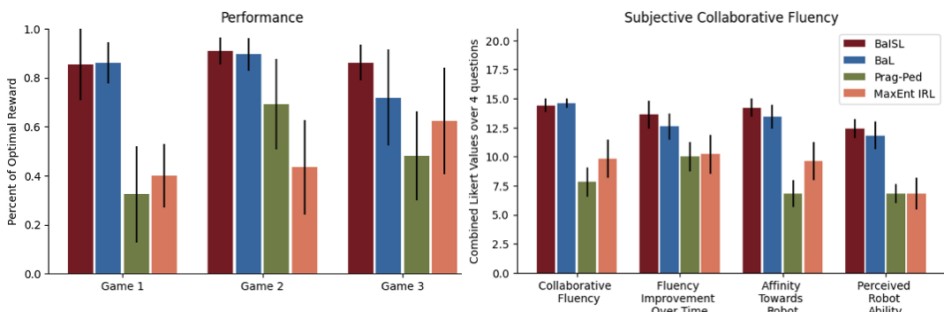

Figure 5: Participants achieved higher percent of optimal reward and higher subjective collaborative fluency with BaISL and BaL over Prag-Ped and MaxEnt IRL. In the second and third rounds, teams with BaISL outperformed BaL in performance.

optimizing for reward, which may result in lower fluency in earlier rounds. Participants rated their affinity towards the robot as a collaborative partner the highest with the BaISL agent on average, and most strongly perceived that BaISL had learned their correct individual reward functions (Figure 5).

## 7 Discussion and Limitations

In this work, we formulate a type of collaborative interaction where the robot must align itself to the human's contribution preferences while adhering to task objectives and its own constraints as an ICPL game. ICPL games are limited to interactions that are based on reward learning, which does not span all types of human-robot collaborations. In ICPL, the robot's constraints are assumed to take the form of a reward function. However, this is not the only form of robot constraint possible; there include activity deadlines, probable errors, and task specifications. Furthermore, this formulation limits the human individual reward as it only captures the notion of the human's preferences for their own contribution, since only the human action is taken as a parameter. However, humans may have preferences or expectations for how they want the robot to contribute. Capturing this contribution preference requires modifying the human reward function to include other agents' actions as input. Despite its limitations, ICPL poses a flexible, interaction paradigm in which robots can represent and learn human contribution preferences in collaborative human-robot tasks.

In our simulated evaluations, we find that, naturally, teams consisting of a rational human partner matching the robot's hypothesized human model yields the highest team rewards. A drawback in testing with simulated human models is that they rely on predefined decision models. Future work involves running larger user studies with human participants, whose decision-making may be more variable and not captured by our predefined models. Another limitation is that our information gain objective does not construct long-horizon multi-state plans towards states with high information potential. Rather, BaISL evaluates whether the next state can be informative and seeks out next states with high information gain potential. BaISL is myopic in its information gain computation, in which the algorithm can only reason about the next state, without a more long-term plan. While this is one limitation of our approach, we believe this also affords the algorithm simplicity and interpretability of its learning process to human partners, demonstrated by the perceived robot's ability to learn in our preliminary user study (Fig. 5). Future work includes active learning methods [53] which consider longer-horizon plans for learning about human partners in collaborative tasks. Lastly, our model is evaluated on a simple task domain for both simulated and real user experiments. While the task includes a coordination complexity in which particular actions must be performed jointly in order for the team to receive reward, it still remains rather simple in the state and action space. Though we include a second task domain (see Appendix), in future work, we plan to scale up task complexity to include more agent interdependencies and preconditons for different actions.

## 8 Conclusion

We introduce a collaborative paradigm, ICPL, where successful teams optimize task objectives while adhering to individual agent constraints. The ICPL framework aims to equip robots with the ability to represent, learn, and accommodate human contribution preferences in collaborative human-robot tasks. We also contribute a candidate approach, BaL, and extension BaISL, which models humans as maximizing rewards with the assumption that the robot will choose the optimal action, facilitating the robot's eventual learning of optimal actions. We evaluated our approach through experiments with simulated human models and a preliminary user study involving human partners.

**Acknowledgements:** This research is supported by the DARPA award FP00002636 and NSF IIS-2112633.

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

# 9 Appendix

## 9.1 Intuitively Understanding Individual Rewards as Individual Costs

**Task Reward** represents the task objective. **Human Individual Reward** represents the human partner's collaborative preferences. And **Robot Individual Reward** represents the robot's soft constraints. The individual rewards are based on the actions each agent individually takes. We can interpret these as negative costs, where an action with lower rewards is straining on the agent performing it. In the dishwasher loading example, picking up an object, like a bowl or glass cup, are determined to be especially costly for the robot because of the difficulty in picking up objects with smooth surfaces with a two finger gripper, and the risk of the dropping and shattering the object. These high cost actions involved in picking up those objects would give low **Robot Individual Reward**.

## 9.2 Bayesian Inverse Reinforcement Learning: Implementation Details

We model the human as a noisily rational actor who selects actions based on the assumption that the robot will take the optimal complementary action to maximize the team's reward. The human implicitly sets expectations for the robot by assuming the robot will take the optimal action. For example, the predicted robot action is optimal under $R_{\theta^T, \theta^r, \theta^h}$, which the human has full knowledge of. By understanding these expectations, the robot is able to deduce which human individual reward function $R_{\theta^h}$ would lead to such expectations. We also adopt the Boltzmann rational model of human behavior [59, 60], which models that human decisions are exponentially likely with respect to reward. According to this model, the probability of choosing a particular option increases exponentially as its utility, or reward, increases compared to other available options.

Over $t$ timesteps, the robot observes the human perform some trajectory of actions, $\tau_t = \{(s_1, a_1^h, s_2), (s_2, a_2^h, s_3), ..., (s_t, a_t^h, s_{t+1})\}$. Since the human has full knowledge of the composite reward, we assume the human's policy is a stationary expert policy. Thus, we can make the following independence assumption for all candidate human reward functions $\hat{\theta}^h \in \Theta^h$ (Equation 10).

$$P(\hat{\theta}^h | \tau_t) \propto P((s_1, a_1^h)|\hat{\theta}^h)P((s_2, a_2^h)|\hat{\theta}^h)...P((s_t, a_t^h)|\hat{\theta}^h)P(\hat{\theta}^h) \tag{10}$$

We intuit that since the human has full knowledge of all three reward functions: task, human individual, and robot individual reward, and in absence of a model of the robot's behavior, the human will try to achieve the task objective while acting pedagogically by taking actions that seek to demonstrate a complementary action the human expected the robot to have performed. This best-case reward, $r_{bc}$, is the reward the team would receive from the composite function $R_{\theta^T, \theta^r, \hat{\theta}^h}$ if the robot had taken the action $a^r \in \mathcal{A}^r$ that would maximize the composite reward (Equation 2). The composite function under candidate $\hat{\theta}^h$ uses $\hat{\theta}^h$ to parameterize the human individual reward term.

$$r_{bc}(s_t, a_t^h | \hat{\theta}^h) = \max_{a^r \in \mathcal{A}^r} \sum_{s_{t+1}} R_{\theta^T, \theta^r, \hat{\theta}^h}(s_t, a_t^h, s_{t+1}) T(s_{t+1}|s, a_t^h, a_t^r)$$

$$P((s_t, a_t^h)|\hat{\theta}^h) \propto \exp(\beta \cdot r_{bc}(s_t, a_t^h | \hat{\theta}^h))$$

Consider a state in which the human has two actions to choose from $a$, and $b$. The robot's evaluates its best-case reward for two hypotheses: (1) best-case reward 1 to $a$, 3 to $b$, and (2) $r_{bc} = 2$ to $a$ and 3 to $b$. The human chooses action $b$; however, we do not want to necessarily update based on a distribution derived from the reward values themselves. Since our robot models the human is a reward maximizer, the human views no effective differences between the two strategies. Thus, we want to ensure the probabilities reflect equal probability for selecting $b$ by thresholding the Boltzmann potential if the action yields a best-case reward equal to the maximum.

$$r(s_t, a_t^h | \hat{\theta}^h) = \begin{cases} \lambda, & \text{if } r_{bc}(s_t, a_t^h | \hat{\theta}^h) = \max_{a^h \in \mathcal{A}^h} r_{bc}(s_t, a^h | \hat{\theta}^h) \\ 1 - \lambda, & \text{otherwise} \end{cases}$$

**Algorithm 1** (BIRL) Bayesian Inverse Reinforcement Learning: Online Update

---

**Input**: State $s_t$, Human Action $a_t^h$, Belief prior $b_0(\hat{\theta}^h) \, \forall \hat{\theta}^h \in \Theta^h$
**Parameter**: Rationality threshold $\lambda$, Hypothesis space $\Theta^h$, Temperature $\beta$
**Output**: Updated beliefs $b$, Predicted human policy $\hat{\pi}^h$

1: **for** $\hat{\theta}^h \in \Theta^h$ **do**
2: $\quad r_{bc}(s_t, a_t^h | \hat{\theta}^h) = \max\limits_{a^r \in \mathcal{A}^r} \sum\limits_{s_{t+1}} R_{\theta^T, \theta^r, \hat{\theta}^h}(s_t, a_t^h, s_{t+1}) T(s_{t+1} | s, a_t^h, a_t^r)$

3: $\quad r(s_t, a_t^h | \hat{\theta}^h) = \begin{cases} \lambda, & \text{if } r_{bc}(s_t, a_t^h | \hat{\theta}^h) = \max\limits_{a^h \in \mathcal{A}^h} r_{bc}(s_t, a^h | \hat{\theta}^h) \\ 1 - \lambda, & \text{otherwise} \end{cases}$

4: $\quad Z_t = \sum\limits_{\theta \in \Theta^h} e^{\beta \cdot r(s_t, a_t^h | \theta)}$

5: $\quad P((s_t, a_t^h) | \hat{\theta}^h) = \frac{1}{Z_t} e^{\beta \cdot r(s_t, a_t^h | \hat{\theta}^h)}$

6: $\quad b(\hat{\theta}^h) \leftarrow P(\hat{\theta}^h | s_t, a_t^h) = \frac{P((s_t, a_t^h) | \hat{\theta}^h)) b_0(\hat{\theta}^h)}{P(s_t, a_t^h)}$

7: **end for**
8: $\hat{\pi}^h(a^h | s; \hat{\theta}^h) \propto e^{\beta \cdot r(s_t, a_t^h | \hat{\theta}^h)}$
9: $\hat{\pi}^h(a^h | s) = \sum\limits_{\hat{\theta}^h} b(\hat{\theta}^h) \hat{\pi}^h(a^h | s; \hat{\theta}^h)$

---

Algorithm 1 below describes the Bayesian inverse reinforcement learning approach. The predicted human policy is the expected policy under the updated beliefs of the robot. We use $\lambda = 0.9$ in our agents for all simulated and human study experiments.

### 9.2.1 Optimistic Information Gain: Implementation Details

In updating its beliefs, the robot assumes that the human will optimistically choose the action that would achieve maximum reward given that the robot takes the ideal action which would facilitate achieving the maximum reward. The robot updates its beliefs $b$ over the possible values of the human's individual reward, $\theta^h$ using a likelihood function $P((s_t, a_t^h) | \hat{\theta}^h)$ built on this assumption. [10] examines a solution to the CIRL problem in which the human teacher is expected to act pedaogically, while the robot learner, aware and expecting this pedagogy, acts practically under its learned beliefs. Research on human pedagogical reasoning demonstrates that when teaching, humans engage in actions aimed at influencing or altering the beliefs of learners [63]. Our assumption that humans will take actions expecting the optimal, complementary robot action interprets the expected human pedagogy as being through setting expectations of the robot.

While the robot uses the composite reward $R_{\theta^T, \theta^r, \hat{\theta}^h}$ to learn from the human's actions, the robot can seek out states that will give the human the opportunity to demonstrate its true composite reward $R_{\theta^T, \theta^r, \theta^h}$. Our key idea is that the robot's actions affect the state, and in turn affect the human's next actions, the robot can leverage this to actively take actions that will lead the team to states in which the human can provide more informative demonstrations. For example, consider the dishwasher unloading task, where the robot and human must collaboratively unload 3 bowls and 1 cup (see Figure 6). If the robot reaches for the cup, it leaves the human with no choice but to unload one of the three remaining bowls. Had the robot reached first for one of the bowls, the human would have the option of choosing between the cup or one of the three bowls. Opting for this more informative state would have given the robot more information about the human's preference between bowls and cups, since the human would have made a decision between the two objects. We will next formalize this desire to maneuver the team into informative state using an information gain metric.

Algorithm 2 represents the planning algorithm which seeks out next states with high potential information gain. In line 5, we perform a Bellman backup to compute a task-based $Q$ given the current beliefs $b$ (Equation 5). $Q(s, a^r, a^h, b)$ represents expected discounted future composite team rewards given $b$. The robot's information gain

$$I(b, s, a^r) = H(b) - \mathbb{E}_{\hat{\pi}^h, T}[H(b | s, a^r)]$$

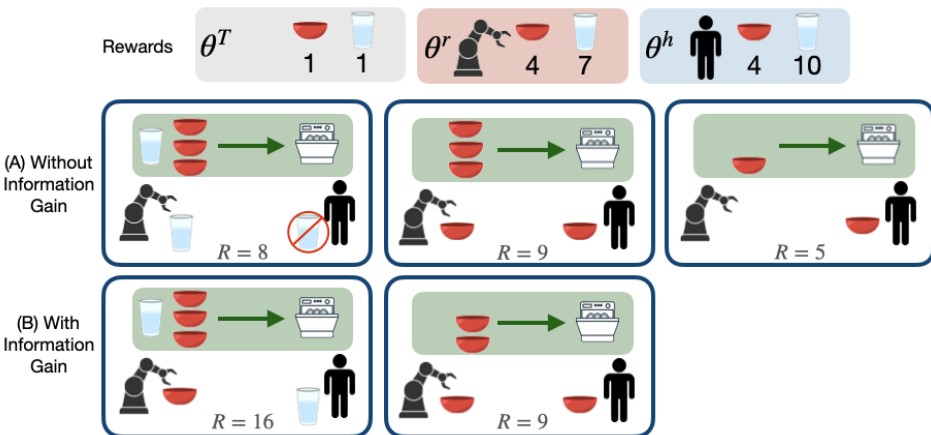

Figure 6: This toy example demonstrates an example of the effect of the information gain measure on the behavior of the robot. The robot and human must collaborate to load a dishwasher with 3 bowls and 1 cup; the team gets +1 reward for loading any object. The robot's individual reward is +7 for the cup, and +4 for the bowl. The human's individual reward is +10 for the cup, and +4 for the bowl. The robot's hypothesis space is two options: (1) +10 for the cup and +4 for the bowl, or (2) +4 for the cup and +10 for the bowl. Without information gain (A), the robot reaches for the cup, but it leaves the human with no choice but to unload one of the three remaining bowls, achieving a reward of $8 + 9 + 5 = 22$. With information gain (B), the robot opts for a more informative next state, where the robot is able to learn the human's preference of cups over bowls, achieving higher reward of $16 + 9 = 25$.

with expected entropy $\mathbb{E}_{\hat{\pi}^h, T}[H(b|s, a^r)]$ under the predicted human policy $\hat{\pi}^h$ and the transition dynamics $T$, using Equation 6. In line 11 of Algorithm 2, the information gain of $a^r$ in state $s$ is added to the Q-value as a boost to actions that would guide the team towards most informative states (Equation 8). If the next state is not informative given any human action, then there will be no information gain boost given to any action. The $\alpha$ term (Equation 9) is a switch that turns on and off the information gain boost. The information gain objective seeks to reduce uncertainty over the human individual reward functions in $\Theta^h$. However, multiple reward functions can lead to the same predicted human action. When the beliefs begin converging on the same predicted human action, the robot will deprioritize seeking information gain. $\alpha$ measures the intersection of the shared predicted human actions by the top probability human individual reward functions in the beliefs $b$.

$$\alpha = | \bigcap_{\hat{\theta}^h \in \Theta^h} \{a \mid \pi^h(a|s; \hat{\theta}^h) = \max_b \hat{\pi}^h(b|s; \hat{\theta}^h) \wedge b(\hat{\theta}^h) = \max_{\bar{\theta}^h} b(\bar{\theta}^h)\}| \tag{11}$$

The robot thus replans under updated beliefs, while seeking informative states in which the human has opportunity to provide informative decision. If the next state is not informative given any human action, then there will be no information gain boost.

Upon observing the action of the human, current state, and next state, the robot updates its beliefs ($b$) over the hypothesis space ($\Theta^h$) and produces a predicted policy for the human ($\hat{\pi}^h$). The robot then replans its actions seeking information gain. This online decision process comprises our full algorithm: Bayesian Information Seeking Learner (BaISL). See Algorithm 3. BaISL is an approach a robot may take to the ICPL task. While the task is not complete, in each timestep, the robot updates its beliefs based on the previous human action and state using Algorithm 1. Then, the robot replans seeking potential information gain using Algorithm 2 (line 8). The robot samples an action from its computed policy (line 9), while the human chooses an action as well. The environment transitions to the next state. Once the task is complete, the taem receives the total composite reward obtained over the interaction (line 14).

For the Bayesian Learner (BaL), we leave out the information seeking term. See Algorithm 4. While the task is not complete, in each timestep, the robot updates its beliefs based on the previous human action and state using Algorithm 1. Then, the robot replans using expected reward under the current

---
**Algorithm 2** (PSIG) Plan Seeking Information Gain
---
**Input**: Beliefs over human reward functions $b$, Predicted human policy $\hat{\pi}^h$
**Parameter**: $\theta^r, \theta^T$
**Output**: $\pi^r$
1: $V(s) \leftarrow$ Initialize $V(s)$ randomly
2: **while** not converged **do**
3:    **for** $s \in \mathcal{S}$ **do**
4:      **for** $(a^r, a^h) \in \mathcal{A}^r \times \mathcal{A}^h$ **do**
5:         $Q(s, a^r, a^h, b) = \sum_{\hat{\theta}^h} b(\hat{\theta}^h) \sum_{s' \in S} T(s'|s, a^r, a^h)\Big(R_{\hat{\theta}^h, \theta^r, \theta^T}(s, a^r, a^h, s') + \gamma V(s')\Big)$
6:      **end for**
7:      $V(s) \leftarrow \max\limits_{(a^r, a^h) \in \mathcal{A}^r \times \mathcal{A}^h} Q(s, a^r, a^h, b)$
8:    **end for**
9: **end while**
10: $\alpha = |\bigcap_{\hat{\theta}^h \in \Theta^h} \{a \mid \pi^h(a|s; \hat{\theta}^h) = \max_b \hat{\pi}^h(b|s; \hat{\theta}^h) \wedge b(\hat{\theta}^h) = \max_{\bar{\theta}^h} b(\bar{\theta}^h)\}|$

11: $\pi^r(s) \leftarrow \arg\max\limits_{a^r \in \mathcal{A}^r} \sum\limits_{a^h \in \mathcal{A}^h} \sum\limits_{\hat{\theta}^h} b(\hat{\theta}^h)\hat{\pi}^h(a^h|s; \hat{\theta}^h)\Big(Q(s, a^r, a^h, b) + \alpha I(b, s, a^r)\Big)$

---

---
**Algorithm 3** (BaISL) Bayesian Information Seeking Learner: An approach to ICPL
---
**Input**: $s_0, \theta^T, \theta^r, \theta^h$
**Output**: Updated beliefs $b$, Predictive model of human action $f$
1: $b \leftarrow$ Initialize uniform prior over $\theta^h \in \Theta^h$
2: **for** task not over **do**
3:    Human observes $\theta^T, \theta^r, \theta^h$
4:    Robot observes $\theta^T, \theta^r$, and $\Theta^h$, but not $\theta^h$
5:    $s \leftarrow s_0$
6:    **while** game $n$ not over **do**
7:      $b, f \leftarrow \text{BIRL}(s, a^h, b)$                     {update beliefs about human utility}
8:      $\pi^r \leftarrow \text{PSIG}(b, f)$               {plan seeking info gain using updated beliefs}
9:      $a^r \sim \pi^r(\cdot|s)$                  {sample robot action $a^r$ from policy}
10:     $a^h \leftarrow$ Human decides $a^h$ based on $s$           {human takes $a^h$}
11:     $s \leftarrow s' \sim T(s'|s, a^h, a^r)$        {environment transitions to state $s'$}
12:    **end while**
13: **end for**
14: Return $R_F = \sum_t R_{\theta^T, \theta^r, \theta^h}(s_t, a_t^r, a_t^h, s_{t+1})$   {team observes final reward once task complete}

---

beliefs using Algorithm 5 (line 8). The robot samples an action from its computed policy (line 9), while the human chooses an action as well.

## 9.3 Simulated Evaluation

### 9.3.1 Simulated Human Models

We experiment with three types of simulated humans. For each agent type, we experiment with temperatures $\beta \to \infty$ (rational), and the other with $\beta = 1$.

1. The *optimistic reward human* selects actions that maximize composite reward assuming the human takes the optimal complementary action. This model is the one used in our BIRL likelihood function, making it easier for *Ours* and *Ours wo IG* to perform well with. The simulated human model selects action $a^h$ with probability proportional to the reward, with a

---
**Algorithm 4** (BaL) Bayesian Learner: An approach to ICPL
---
**Input**: $s_0, \theta^T, \theta^r, \theta^h$
**Output**: Updated beliefs $b$, Predictive model of human action $f$
 1: $b \leftarrow$ Initialize uniform prior over $\theta^h \in \Theta^h$
 2: **for** task not over **do**
 3:     Human observes $\theta^T, \theta^r, \theta^h$
 4:     Robot observes $\theta^T, \theta^r$, and $\Theta^h$, but not $\theta^h$
 5:     $s \leftarrow s_0$
 6:     **while** game $n$ not over **do**
 7:         $b, f \leftarrow \text{BIRL}(s, a^h, b)$              {update beliefs about human utility}
 8:         $\pi^r \leftarrow \text{PLAN}(b, f)$              {plan seeking info gain using updated beliefs}
 9:         $a^r \sim \pi^r(\cdot|s)$              {sample robot action $a^r$ from policy}
10:         $a^h \leftarrow$ Human decides $a^h$ based on $s$              {human takes $a^h$}
11:         $s \leftarrow s' \sim T(s'|s, a^h, a^r)$              {environment transitions to state $s'$}
12:     **end while**
13: **end for**
14: Return $R_F = \sum_t R_{\theta^T, \theta^r, \theta^h}(s_t, a_t^r, a_t^h, s_{t+1})$   {team observes final reward once task complete}

---
**Algorithm 5** (PLAN) Plan Under Expected Rewards Given Current Beliefs
---
**Input**: Beliefs over human reward functions $b$, Predicted human policy $\hat{\pi}^h$
**Parameter**: $\theta^r, \theta^T$
**Output**: $\pi^r$
 1: $V(s) \leftarrow$ Initialize $V(s)$ randomly
 2: **while** not converged **do**
 3:     **for** $s \in \mathcal{S}$ **do**
 4:         **for** $(a^r, a^h) \in \mathcal{A}^r \times \mathcal{A}^h$ **do**
 5:
$$Q(s, a^r, a^h, b) = \sum_{\hat{\theta}^h} b(\hat{\theta}^h) \sum_{s' \in S} T(s'|s, a^r, a^h) \left( R_{\hat{\theta}^h, \theta^r, \theta^T}(s, a^r, a^h, s') + \gamma V(s') \right)$$
 6:         **end for**
 7:         $V(s) \leftarrow \max_{(a^r, a^h) \in \mathcal{A}^r \times \mathcal{A}^h} Q(s, a^r, a^h, b)$
 8:     **end for**
 9: **end while**
10: $\pi^r(s) \leftarrow \arg\max_{a^r \in \mathcal{A}^r} \sum_{a^h \in \mathcal{A}^h} \sum_{\hat{\theta}^h} b(\hat{\theta}^h) \hat{\pi}^h(a^h|s; \hat{\theta}^h) \left( Q(s, a^r, a^h, b) \right)$

---

best-case prediction for $a^r$.

$$a^r \sim \pi^h(a^h|s_t) \propto exp\left( \beta \max_{a^r \in \mathcal{A}^r} \sum_{s_{t+1}} R_{\theta^T, \theta^r, \hat{\theta}^h}(s_t, a_t^h, s_{t+1}) T(s_{t+1}|s, a_t^h, a_t^r) \right)$$

2. The *optimistic reward human* selects actions that maximize *expected* composite reward under a uniform probability over all robot actions. This human decision making function does not reflect Prag-Ped nor Ours, making this human model out of distribution for both. The human model computes its reward by marginalizing out $a^r$.

$$\pi^h(a^h|s_t) \propto exp\left( \beta \sum_{a^r \in \mathcal{A}^r} \sum_{s_{t+1}} R_{\theta^T, \theta^r, \hat{\theta}^h}(s_t, a_t^h, s_{t+1}) T(s_{t+1}|s, a_t^h, a_t^r) \right)$$

3. The *pedagogic human* uses the Pragmatic-Pedagogic Value Alignment [10] Q-value corresponding to the true human individual reward. Since the human computes the expected Q-value for its own actions by marginalizing over robot actions. This model is the one used in the *Prag-Ped* robot's likelihood function, making it easier for *Prag-Ped* to work well with. The Pragmatic-Pedagogic Value Alignment solution to CIRL leverages the assumption that the human can observe the robot's action at the current timestep before selecting its own action. The

human policy maximizes the best expected outcome for each available action:

$$\pi^h(a^h|s,b,a^r,\hat{\theta}^h) \propto exp(\beta Q(s,b,a^h,a^r;\hat{\theta}^h)$$

In order to compute $Q$, the human considers the belief update of the robot, where the update of the robot's belief is determine given by the Bayesian update:

$$b'(\hat{\theta}^h|s,b,a^r,a^h) \propto \pi^h(a^h|s,b,a^r,\hat{\theta}^h)b(\hat{\theta}^h)$$

The robot's policy under the new beliefs maximizes the expected $Q$ under the new beliefs.

$$\pi^{r*}(s',b') = \arg\max_{a^r} \sum_{a^h,\hat{\theta}^h} Q(s,b,a^h,a^r;\hat{\theta}^h)b(\hat{\theta}^h)$$

The Bellman equation for the human is as follows:

$$Q(s,b,a^h,a^r;\hat{\theta}^h) = R_{\theta^T,\theta^r,\hat{\theta}^h}(s,a^r,a^h) + \mathbb{E}_{s',a^{h'}}\left[\gamma Q'(s',b',a^{h'},\pi^{r*}(s',b');\hat{\theta}^h)\right]$$

The human is pedagogic because the Bellman equation takes into account how the robot's beliefs will change based on the actions of the human. Since the human cannot actually see the robot's action ahead of time during this collaborative decluttering task, the human marginalizes out $a^r$:

$$\pi^h(a^h|s,b,a^r,\hat{\theta}^h) \propto exp(\sum_{a^r \in \mathcal{A}^r} \beta Q(s,b,a^h,a^r;\theta^h))$$

and acts according this policy.

### 9.3.2 Approaches

1. **BaL: Bayesian Learner** The robot policy $\pi^r$ is defined by Equation 8, but without the information gain term. The robot models an *optimistic reward human* and plans using expected reward under current beliefs. The robot selects the action $a^r$ maximizing the expected-reward-based Q-values only. The robot selects the action $a^r$, according to Algorithm 4, maximizing the expected-reward-based Q-values.

$$\pi^r(s) \leftarrow \max_{a^r \in \mathcal{A}^r} \sum_{a^h \in \mathcal{A}^h} \sum_{\hat{\theta}^h} b(\hat{\theta}^h)\hat{\pi}^h(a^h|s;\hat{\theta}^h)\left(Q(s,a^r,a^h,b)\right)$$

2. **BaISL: Bayesian Information Seeking Learner** We consider an extension to BaL, where the robot plans towards states that have high information gain potential. The robot acts according to $\pi^r$ (Eq. 8), modeling the human as an optimistic reward teacher and planning towards informative states. The robot selects the action $a^r$, according to Algorithm 3, maximizing the expected-reward-based Q-values and information gain boost.

$$\pi^r(s) \leftarrow \max_{a^r \in \mathcal{A}^r} \sum_{a^h \in \mathcal{A}^h} \sum_{\hat{\theta}^h} b(\hat{\theta}^h)\hat{\pi}^h(a^h|s;\hat{\theta}^h)\left(Q(s,a^r,a^h,b) + \alpha I(b,s,a^r)\right)$$

3. **Prag-Ped: Pragmatic-Pedagogic Value Alignment [10]** As a baseline, we compare the performance of our agent against a solution to the CIRL [12] problem in which the human acts pedagogically while the robot reasons practically. The robot policy assumes that the human will act pedagogically with a Q value function that accounts for the robot's beliefs. The Pragmatic-Pedagogic value alignment solution further assumes the human observes the robot's action before selecting their own action. The human policy maximizes the best expected outcome for each available action:

$$\pi^h(a^h|s,b,a^r,\hat{\theta}^h) \propto exp(\beta Q(s,b,a^h,a^r;\hat{\theta}^h)$$

Consider a state in which the human has only one action. an incorrect hypothesize reward achieved will be higher for , since under Thus, we want to ensure the probabilities reflect

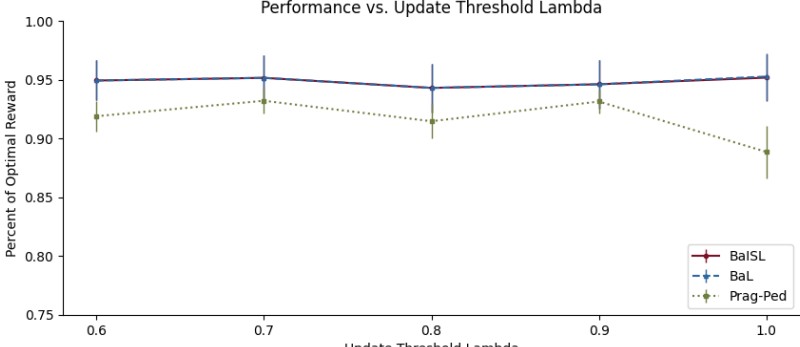

Figure 7: Performance evaluation of our models across different values of our $\lambda$ hyperparameter used in BIRL (line 3, Algorithm 1) shows that the **BaISL and BaL** methods are unaffected by $\lambda$ when partnering with the $\beta = 1$ *optimistic reward human*.

equal probability for selecting $b$ by thresholding the Boltzmann potential if the action yields a best-case reward equal to the maximum.

$$r(s_t, a_t^h | \hat{\theta}^h) = \begin{cases} \lambda, & \text{if } r_{bc}(s_t, a_t^h | \hat{\theta}^h) = \max_{a^h \in \mathcal{A}^h} r_{bc}(s_t, a^h | \hat{\theta}^h) \\ 1 - \lambda, & \text{otherwise} \end{cases}$$

In order to compute $Q$, the human considers the belief update of the robot, where the update of the robot's belief is determine given by the Bayesian update:

$$b'(\hat{\theta}^h | s, b, a^r, a^h) \propto \pi^h(a^h | s, b, a^r, \hat{\theta}^h) b(\hat{\theta}^h)$$

The robot's policy under the new beliefs maximizes the expected $Q$ under the new beliefs.

$$\pi^{r*}(s', b') = \arg\max_{a^r} \sum_{a^h, \hat{\theta}^h} Q(s, b, a^h, a^r; \hat{\theta}^h) b(\hat{\theta}^h)$$

The Bellman equation for the human is as follows:

$$Q(s, b, a^h, a^r; \hat{\theta}^h) = R_{\theta^T, \theta^r, \hat{\theta}^h}(s, a^r, a^h) + \mathbb{E}_{s', a^{h'}} \left[ \gamma Q'(s', b', a^{h'}, \pi^{r*}(s', b'); \hat{\theta}^h) \right]$$

While the Prag-Ped solution computes both a policy for the human and robot, we take and execute the policy of the robot for this robot baseline. The robot takes the action maximizing expected reward under the beliefs, using $\pi^{r*}(s', b')$. The *pedagogic human* simulated model acts according to the human policy part of the Prag-Ped solution.

4. **MaxEnt** Our second baseline is Maximum Entropy Invese Reinforcement Learning [50], followed by replanning using the learned reward. The robot learns a reward function $f(s, a^r, a^h)$ representing $R_{\theta^T, \theta^r, \hat{\theta}^h}(s, a^r, a^h)$ using the demonstrated states and joint actions previously seen. The robot evaluates Q-values based on the learned reward function $f$:

$$Q(s, a^r, a^h; f) = f(s, a^r, a^h, s') + \gamma \sum_{s' \in S} T(s' | s, a^r, a^h) \max_{a^{r'}, a^{h'}} Q(s', a^{r'}, a^{h'}; f)$$

$$\pi^{r*}(s) = \arg\max_{a^r} \sum_{a^h} Q(s, a^r, a^h; f)$$

### 9.3.3 Hyperparameter Analysis

We analyze the performance of our models across the $\lambda$ hyperparameter used in BIRL (line 3, Algorithm 1). We test $\lambda$ from 0.6 to 1.0 in 0.1 intervals, using the $\beta = 1$ *optimistic reward human* with robots **BaISL, BaL, Prag-Ped**, using 50 random game configurations. We find that performance

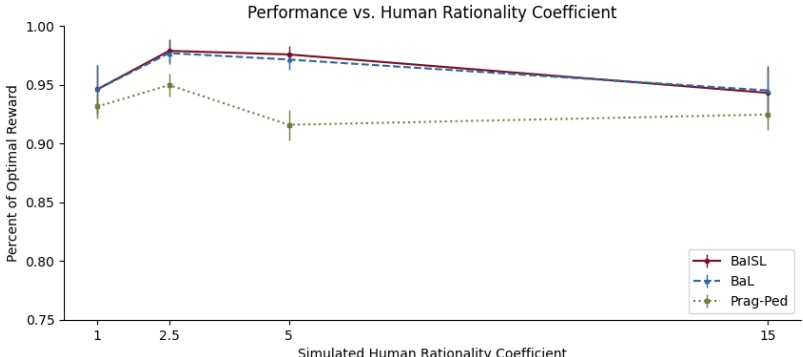

Figure 8: We find that performances of **BaISL, BaL**, and **Prag-Ped** are not affected by true $\beta$ value for the simulated human.

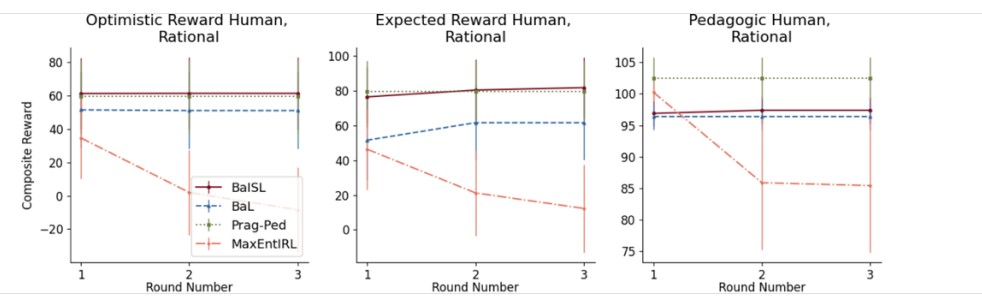

Figure 9: Evaluation with Rational Human Models in Team Space Fortress task. We evaluate the agents across 3 rounds of interaction with the same 3 human models used in the decluttering task. Across all 3 models, the BaISL agent reliably outperforms the BaL and MaxEntIRL models. BaISL and Prag-Ped perform similarly on the *optimistic reward human* and *expected reward human*. Prag-Ped outperforms all 3 other methods when paired with the Pedagogic human model. The error bars represent standard error.

of our algorithm BaISL is not affected by the choice of $\lambda$ value. The **Prag-Ped** robot is mostly unaffected as well, but performance drops slightly with $\lambda = 1.0$. We use $\lambda = 0.9$ in our agents for all other simulated experiments. Additionally, we analyze the performance of our models across the $\beta$ hyperparameter used in the simulated human model for the *optimistic reward human* with robots **BaISL, BaL, Prag-Ped**, using 50 random game configurations. The robots update their beliefs using a beta of 1, while the TRUE $\beta$ values for the simulated human are plotted on the x-axis in Figure 8. We find that performances of **BaISL, BaL**, and **Prag-Ped** are not affected by true $\beta$ value for the simulated human.

## 10    Additional Domain: Team space fortress (TSF)

Team space fortress (TSF) is an abstracted variant of the task in Li et al. [64]. The human and robot team must defeat an enemy spaceship. To do so, one agent must act as bait, while the other shoots the enemy. Unknown to robot, the human prefers to serve as either the bait or the shooter. The robot's constraints represent a proficiency to either shoot or bait as well. The action space is $\mathcal{A}^h = \mathcal{A}^r = \{act\ as\ bait, position\ to\ bait, wait, position\ to\ shoot, shoot\}$. This task introduces preconditions, which are dependent on prior actions and the agent who performed such actions. For instance, in order to shoot the enemy, an agent must have positioned themselves to shoot. Agents must *position to bait* before they *act as bait*. An agent cannot act as bait if the other agent has already positioned themselves to bait, and similarly for the shooter. Compared to the initial table decluttering task, this task introduces several complexities: history-dependent state preconditions for actions, a larger action space, and a larger state space.

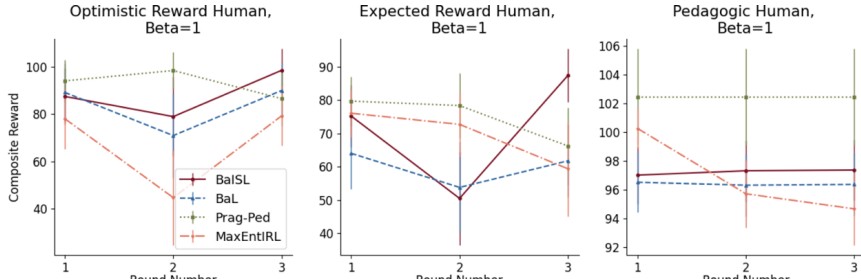

Figure 10: Evaluation with $\beta = 1$ Human Models in Team Space Fortress task. We evaluate the agents across 3 rounds of interaction with the same 3 human models used in the decluttering task. After the 3 interactions, for the Optimistic Reward and Expected Reward humans, the BaISL agent outperforms the BaL model. Prag-Ped outperforms the other models when paired with the Pedagogic human model and generally outperforms the others on the *optimistic reward human* and *expected reward human* in the first two rounds.

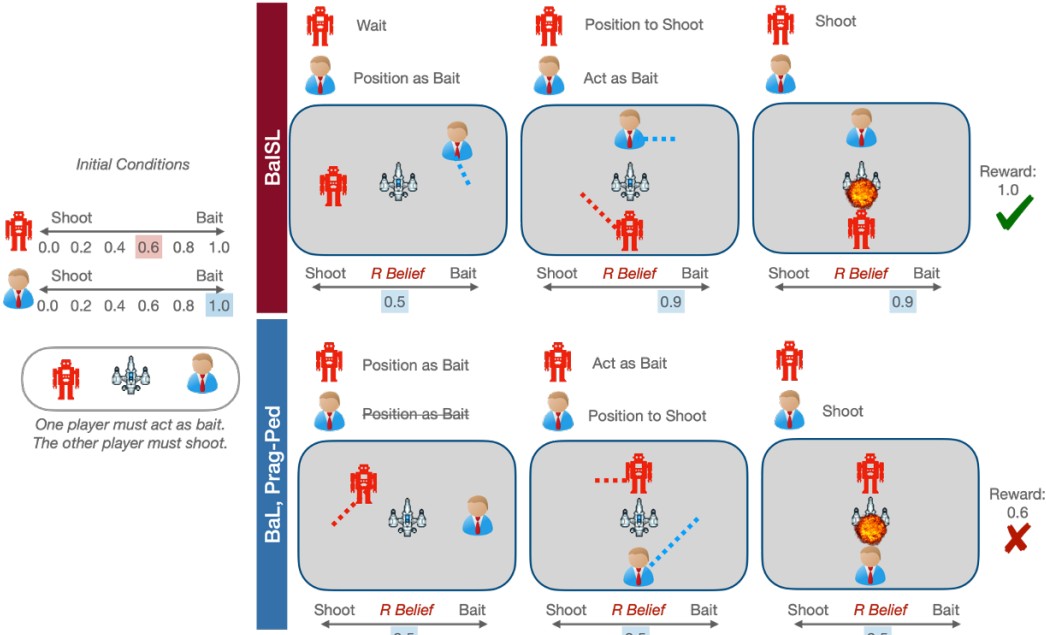

Figure 11: Illustration of BaISL in Team Space Fortress task. Initially, the human has a strong preference for acting as the bait, while the robot has only a slight preference for acting as bait. The BaISL robot (top row) selects to wait on the first turn, and observes the human position themselves as bait. The robot is able to update its belief of the human's preference, denoted R belief. On the other hand, the BaL and Prag-Ped robot (bottom row) goes straight for acting as bait, which prohibits the human from doing so as well. The human has no choice but to act as the shooter, which does not enable the robot to learn the human's reward. Thus, the BaISL robot is able to achieve optimal reward.

To evaluate, we randomly sample the reward values for the individual constraints of each agent. In the TSF environment, we demonstrate that our method outperforms the baseline methods, over 20 random configurations (see Figure 9 & 10). In particular, BaISL consistently outperforms BaL and MaxEntIRL across all human models. BaISL performs similarly to Prag-Ped on the Optimistic Reward and Expected Reward humans, and Prag-Ped consistently outperforms the others when paired with the *pedagogic human*. In Figure 11, we illustrate a particular instance of the TSF task in which the BaISL robot seeks to learn before diving into the task, demonstrating the impact of the information gain term.

## 10.1 Elaboration on Viewing Human and Robot Constraints as Separate Reward Functions

Since the human knows the robot constraint, we in theory could learn 'human constraint+robot constraint' as one reward function for the human, which the robot will seek to learn. Viewing the two as separate allows for each constraint to represent conceptually distinct preferences. For example, the robot's constraint $R_{\theta^r}(s, a_r, s')$ could be a temporal constraint, where $s$ includes time. The human's constraint may not involve time, so $s$ may be featurized or represented differently.

Additionally, viewing the constraints as separate allows for the robot to evaluate the human's preferences and transfer the knowledge to other robots with potentially different constraints. For instance, when interacting with robots with different capabilities, the underlying human preferences don't change, but the human's policy, which includes robot constraints, may result in different actions.

Separating the two also allows for this framework to be more generalized when we take knowledge into account. The representations of individual constraints in the collaborative context can be easily used to allow for situations where the human does not know the robot's constraints.

