# OpenReview forum: "Learning Human Contribution Preferences in Collaborative Human-Robot Tasks"
_robot-learning.org/CoRL/2023/Conference — CoRL 2023 Poster_

### Official Review · Reviewer_92c8 · 2023-07-19

**Confidence:** 3
**Originality:** Good
**Technical Quality:** Very Good
**Clarity Of Presentation:** Very Good
**Impact:** 3

**Recommendation:**

Weak Accept: I recommend accepting the paper, but will not argue for my recommendation if the majority of other reviewers have a different opinion.

**Review:**

The formulation used in the paper is, for the most part, sensible.  From a problem formulation point-of-view, it seems sensible to assume there are situations in which the human has complete observability of the task, the robot’s constraints, and their own preferences, while the robot has complete observability of all but the human’s preferences.  The Bayesian inference approach seems sound in the context of the problem setup.  This is common to many papers dealing with information gain, and use of information to improve the legibility of one of the player’s contributions as observed by another player.

In the evaluation, use of the three simulated human actors are well-motivated.  The baselines are also well-selected, and the results show improvement over these baselines.  However, it would be interesting to see more evaluations with other realistic types of humans, including those that do not follow Boltzmann rationality, and those that are adversarial, choosing actions that degrade the task.

The primary concern with work is that the authors only evaluate in simplistic task scenarios with only high-level actions considered (block clean up task with three object types).  The action space is extremely simplistic and the transition function is trivial.  It is difficult to assess the scalability of the approach, and understand precisely how it may handle continuous tasks that are of higher complexity.  The authors should aim to evaluate on another task with larger action space, richer set of rewards and continuous reasoning.  It is also puzzling why only three rounds of interaction were evaluated, and not more.  It would be informative to see the asymptotic behavior of repeated interactions.

The writing in the various sections of the paper could also be improved.  The problem setup in section 3 is quite vague as written, and could benefit from a running example to ground the nomenclature and factorization of the reward structure.  Additionally, it is unclear, without more detail, the motivation behind the information gain switch term in eq. 8 and what motivates the choice.  Is this empirically motivated, or are there theoretical properties?

Minor comments:
The authors should fix all  typos, e.g. “that is robot is able to take” in the definition of the robot individual reward, etc.  The authors should also fix the broken links to equation references in the appendix.

**Quality Of The Limitations Section:**

Limitations are addressed clearly

**Questions For Rebuttal:**

In the Pedagogic Human evaluation, could the authors please comment why the BaISL / BaL approaches do not outperform (or very marginally outperforms) the Prag-Ped approach?

Could the authors comment on the scalability of their approach, and how they may apply the Bayesian approach in the continuous setting?

**Robotics Focus:**

Relevant but unlikely to deploy to hardware in near future

**Summary Of Paper:**

Optimizing for preferences in human contribution in a human-robot collaborative task requires inferring the human’s preferences and goals to optimize the robot’s contribution in the task.  This paper addresses this by contributing: (1) a Markov game representation for human and robot collaboration, (2) a Bayesian inference procedure for active learning the human’s preferences at runtime, and (3) evaluation of the procedure on simulated human partners and a pilot with human users in a manipulation task.

**Summary Of Recommendation:**

Overall, I believe this paper offers sufficient novelty in the Bayesian information-seeking approach and clear improvements over baselines in shared task setting.  As stated above, I would like to see more extensive evaluations in more than one task setting, if possible one with higher complexity, as it is difficult to assess the approach.  I therefore currently rate as weak accept.

---

### Official Review · Reviewer_cJgK · 2023-07-19

**Confidence:** 3
**Originality:** Very Good
**Technical Quality:** Very Good
**Clarity Of Presentation:** Very Good
**Impact:** 3

**Recommendation:**

Weak Reject: I recommend rejecting the paper, but will not argue for my recommendation if the majority of other reviewers have a different opinion.

**Review:**

Strengths:
The paper is well-written and easy to understand. The use of Bayesian IRL to derive beliefs about human rewards and incorporating these beliefs into the planning process is intuitive and logical.
The problem formulation is clearly presented.

Weaknesses:
One potential concern lies in the technical contribution of the paper. While both Bayesian IRL and planning under beliefs are widely used techniques, the authors introduce an entropy term into the planning objective, representing the entropy deduction based on the assumption of humans taking optimal actions in the next time step. However, the impact of this entropy term on the robot's collaborative policy performance appears to be marginal, as observed in the comparison of BaL and BaISL in Figure 2, 3, and 5. Clarification from the authors is needed to understand the technical novelty and significance of this contribution.

Furthermore, the simplicity of the Markov Decision Process (MDP) used in the evaluation is a potential limitation. With a small action space and a short horizon (T=3), more complex evaluation scenarios would be beneficial to demonstrate the effectiveness and robustness of the proposed method.


**Quality Of The Limitations Section:**

Limitations are addressed clearly

**Questions For Rebuttal:**

1. Clarification of the technical novelty of the work
2. More complicated evaluation tasks in addition to the simple MDP used in the current manuscript.

**Robotics Focus:**

Relevant but unlikely to deploy to hardware in near future

**Summary Of Paper:**

The paper under review proposes a Bayesian framework for learning collaborative robot policy. The framework consists of two stages: Bayesian Inverse Reinforcement Learning (IRL) and planning based on the obtained beliefs. The authors evaluate their approach through simulations of collaborative tasks and a user study.

**Summary Of Recommendation:**

Overall, the paper exhibits clear presentation and logical flow. However, to strengthen the technical contribution, a more substantial and distinctive advantage over existing methods should be demonstrated, potentially through more intricate evaluation scenarios that more closely resemble real-world collaborative tasks. I'm leaning toward rejecting the paper at the current moment but would happy to change my score if the authors provide good explanations regarding the technical novelty and more comprehensive experimental results.

---

### Official Review · Reviewer_TiNh · 2023-07-20

**Confidence:** 4
**Originality:** Very Good
**Technical Quality:** Very Good
**Clarity Of Presentation:** Very Good
**Impact:** 3

**Recommendation:**

Weak Accept: I recommend accepting the paper, but will not argue for my recommendation if the majority of other reviewers have a different opinion.

**Review:**

Strengths:

•	The paper is well motivated in such that it considers a new interaction setting in which both human contribution constraints and robot contribution constraints are considered.

•	The experimental evaluation includes both simulated human and real human users, and samples 100 different reward configuration for simulation experiments. The authors conduct user study.

•	The paper is clearly written in detail and the problem is clearly defined.

Weaknesses:

•	The experiment is conducted in only one task setting for both simulated human and real human. In addition, the task is cleaning up three objects, which is not complex as the paper admitted.


**Quality Of The Limitations Section:**

Limitations are addressed clearly

**Questions For Rebuttal:**

In paper’s example of unloading dishwasher, the authors mention human constraints is: prefer the robot to unload non-fragile item, and robot constraints is: it can safely unload the plates. However, it seems the robot constraint can also be viewed as another form of human constraint: the human does not trust robot to unload fragile item but knows the robot can comfortably handle plate’s upright position. What would be the difference of viewing them separately as ‘human constraint +robot constraint’ versus a more strict ‘human constraint’ since human knows robot constraint? What would be some examples that these two could be clearly divided? What would be the reason of separating them if there are no clear differences?

**Robotics Focus:**

Highly relevant to robotics but no hardware experiments

**Summary Of Paper:**

This paper introduces a new form of human-robot interaction, Interactive Contribution Preference Learning (ICPL), that brings human constraints, robot constraints, and their common objectives into consideration. The paper also proposes an approach, Bayesian Information Seeking Learner (BaISL), to solve this new form of task. The method uses a human model that assume robot will choose optimal actions. The robot updates its belief and predict human policy based on observation of human action, and then plan its action aiming for information gain. The authors conduct experiments with both simulated human and real human with promising results.

**Summary Of Recommendation:**

The paper proposes to define and solve human-robot collaboration in which human and robot contribution constraints are considered. The paper is clearly written, well-motivated, and technical sound. The literature review is thorough. The experiment evaluation is simple, but it could be a starting point.

---

### Official Review · Reviewer_Fwxy · 2023-07-24

**Confidence:** 3
**Originality:** Very Good
**Technical Quality:** Very Good
**Clarity Of Presentation:** Good
**Impact:** 4

**Recommendation:**

Weak Accept: I recommend accepting the paper, but will not argue for my recommendation if the majority of other reviewers have a different opinion.

**Review:**

**Strengths**
- Interesting, novel approach to model human-robot collaboration
- Experiments with real humans
- Baselines include ablations and prior work

**Weaknesses**
- Method not described clearly and the intuitive understanding of the advantage over IRL is not well-defined.
- Experiments do not include real robot.

**Detailed Comments**
- Related work: can discuss more work in information gathering and contrast with approaches used in the past.
- Section 4 is not easy to understand from 170. Apart from improving the flow of information, authors can address the following.
  - A number of symbols are mentioned but not defined. E.g., $V$, $f$.
  - Summations sometimes have set mentioned but not always. E.g. $s' \in S$ is there but $\hat{\theta}^h \in$?.
  - Equation 5. What is the expectation over? Not sure what $f, T$ are referring to here.
  - Paragraph 192 is very difficult follow. I'm also not sure how the human is expected to maximize the robot's information gain when it also optimistically assumes the robot to know $\theta^h$.
  - line 211. No-op not defined.

**Quality Of The Limitations Section:**

Limitations are addressed clearly

**Questions For Rebuttal:**

Please refer to review above.

**Robotics Focus:**

Highly relevant to robotics but no hardware experiments

**Summary Of Paper:**

The paper models human-robot collaboration as a two-agent game where the agents share rewards but the full reward is only known to the human. They introduce a Bayesian approach to infer the human-known component of the reward by explicitly including information gain in the robot's policy objective. They evaluate this method in simulation with both simulated and real humans and demonstrate improved team performance.

**Summary Of Recommendation:**

The paper tackles an important problem and introduces an interesting method to model human-robot interaction. The evaluation domain and the baselines are appropriate. I would like the authors to make parts of the paper clearer.

---

### Author Response · Authors · 2023-08-16
**Summary**

As we near the end of the rebuttal period, the authors would like to thank the reviewers for providing valuable and insightful feedback, questions, and suggestions. We appreciated that the reviewers found our framework for human-robot collaboration interesting and original. The primary questions shared by the reviews were regarding:

    (1) clarification of the algorithms,

    (2) better contextualizing the approach within related work,

    (3) conducting experiments on more complex environments.

In our responses:

    (1) We clarified the questions reviewers had about the algorithm, and will incorporate those clarifications into refining our final manuscript.

    (2) We provided additional related works which use information seeking in the robot’s plan, which contextualize our approach within relevant literature.

    (3) We conducted additional experiments and demonstrated performance improvement on a new task, a bait-and-shooter game used in prior literature. The task introduces several complexities: preconditions for actions, a larger action space, and a larger state space.

We believe we have satisfactorily addressed the reviewers’ concerns with our responses, and thank them for their feedback.

---

### Decision · Program_Chairs · 2023-08-30

**Decision:**

Accept (Poster)

**Comment:**

The paper proposes a method for robots to infer human preferences via interaction as a reward function. Three reviewers voted to accept the paper and found the literature review to be thorough, the technical contribution to be solid, but raised issues around experimental evaluation. cJgK voted to reject the paper based on weak experimental results. In their rebuttal, authors included an additional domain which other reviewers appreciated, but cJgK didn't respond despite several messages. I believe the authors have sufficiently addressed the concerns raised by the reviewer. As such, I recommend the paper be accepted.

Some thoughts for the final revision:
-  Please include a discussion about continuous action space / include experiments showing the applicability of method in these domains. - - Clarify the technical contributions as done in the response to cJgK and also suggested by Fwxy.
- Discussion about comaprison with Prag-Ped in response to 92c8 is insightful and should be included in the paper.
- A pedagogical example in the paper is a scenario is where in dishwasher unloading task, the human picks up fragile items whereas the robot picks up silverware/plates. In such scenarios, it seems plausible that the human can express the preferences to the robot in natural language or in formal language. Why should the intent be inferred based on interactions? A discussion on when it is appropriate to directly communicate preferences, v/s inference based on interactions would be very meaningful and help to better contextualize the work.